# Stative vs. Eventive Participles in an Arbëresh Variety under the Influence of the Italian Language

**Giuseppina Turano** 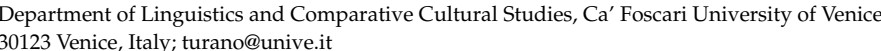

Department of Linguistics and Comparative Cultural Studies, Ca' Foscari University of Venice, 30123 Venice, Italy; turano@unive.it

**Abstract:** In this paper, I explore the properties and the uses of the past participles in the Arbëresh variety of S. Nicola dell'Alto, an Albanian dialect still spoken in Southern Italy, which has been in contact with Italo-Romance varieties for more than five centuries. The data are discussed in comparison to standard Albanian and the contact language, Italian. In Albanian grammar, there is only one type of participle: the past participle. It has both verbal and adjectival properties. As a verbal form, the participle is used in compound and in periphrastic tenses, in combination with both the auxiliaries KAM 'have' and JAM 'be'. It can also be used in combination with other particles to create non-finite verbal forms such as gerund or infinitive or to build up temporal expressions. Finally, it can also be used after some modal impersonal verbs. Verbal participles never show agreement. Albanian participles can also be adjectival. All the adjectives derived by a participial verb take a linking article and always agree with the noun they modify in gender, number, Case and definiteness. The formal distinction of the verbal participles from adjectival participles seems to correlate with the aspectual properties of the construction: a verbal participle appears in eventive structures, whereas an adjectival participle occurs in stative structures. But, as we shall see, this is not always the case. Arbëresh participles have maintained the same morphological and syntactical properties of Albanian. They can be used both in stative and in eventive contexts, but in Arbëresh eventive passives, which are built up as in Italian rather than as in Albanian, the adjectival participles are always inflected. Agreement is obligatory in all the contexts where it is in Italian. This is a clear contact-induced change. The data presented in this paper show that Arbëresh, on the one hand, preserves features of Albanian grammar, whereas, on the other hand, it has undergone changes under the influence of the surrounding Italo-Romance varieties.

**Keywords:** participles; Arbëresh; Albanian; eventive; stative

## 1. Introduction

In this paper, I analyse the properties and the uses of the past participle in eventive and stative contexts in the Arbëresh variety of S. Nicola dell'Alto, an Albanian dialect still spoken in Southern Italy which has been in contact with Italian and Italo-Romance varieties for more than five centuries. The data are examined in comparison to the equivalent data of standard Albanian (to which Arbëresh varieties are genetically related), and to Italian (a typologically distinct language), which is spoken by the entire community. Bilingualism is widespread among Arbëresh speakers. The use of two distinct languages within the same community has led to a heavy diglossia: each language is confined to a particular type of social interaction but with frequent patterns of code alternation and code switching. Arbëresh is used in informal contexts, such as the home and the local neighbourhood, while Italian is used in more formal contexts such as school, culture, administration, and church.

The impact of the language contact is mainly observed in the lexicon where the loanwords from Italian are particularly visible, but we can also find traces in the morphology and in the syntax. So, Arbëresh dialects, after many centuries of contact with Italian, have absorbed traits that are typical of this language.

The data considered here reveal three points of difference between Arbëresh and standard Albanian and similarities between Arbëresh and Italian: (1) In Arbëresh, unlike in standard Albanian, eventive passives are adjectival since they always display obligatory agreement on the lexical participle. (2) In Arbëresh, statives are always adjectival, like in Italian, whereas in standard Albanian, they can also be verbal. (3) The Arbëresh variety considered here, while retaining the same strategies found in Albanian for the realisation of non-active morphology, differs from Albanian and draws closer to Italian in the formation of passive sentences. So, Arbëresh, on the one hand, preserves features of Albanian grammar, while, on the other hand, has undergone changes under the influence of the surrounding Italo-Romance varieties.

This paper is structured as the following: Section 2 illustrates the distribution of verbal participles in Arbëresh, in Albanian, and in Italian. Adjectival participles are introduced in Section 3. Section 4 shows the differences between stative and eventive participles in standard Albanian. Section 5 shows the distribution of eventive and stative participles in Arbëresh, in comparison with Albanian and Italian. Section 6 deals with the structure of stative participles. Section 7 illustrates the differences between Arbëresh and Albanian passives and shows how Italian has played a role in the realisation of Arbëresh passive sentences.

## 2. Verbal Participles in Arbëresh/Albanian in Comparison with Italian

There is a long tradition in the literature that divides participles into two categories, i.e., verbal (1a) and adjectival (1b) passive participles:

(1)  a.  The door was opened
     b.  The door was open

A series of tests has been used to distinguish these two categories (Wasow 1977; Williams 1981; Bresnan 1982; Levin and Rappaport 1986; Kratzer 2000; Embick 2004). For example, adjectival participles can appear as complements of verbs such as *become* or *seem*, while verbal participles cannot (Wasow 1977, p. 339).

(2)  a.  The door seems open
     b.  *The door seems opened

Adjectival participles, unlike verbal ones, can occur after verbs of creation, such as *build, create, make* (Embick 2004, p. 357).

(3)  a.  This door was built open
     b.  *This door was built opened

Adjectival participles, unlike verbal participles, do not allow agentive by-phrases:

(4)  a.  *The door was open by John
     b.  The door was opened by John

Adjectival participles can appear in a prenominal position (Wasow 1977, p. 338):

(5)  The broken box sat on the table

Another test used to distinguish these two types of participles is their ability to support un-prefixation. It seems that verbal participles can take the prefix un-, while un-prefixation is more restricted with adjectival participles:

(6)  a.  An unopened door
     b.  *An unopen door

All these examples show that some participles are adjectives, while other are verbs.

In some languages, participles agree with the subject or the object of the clause, while in other languages, no agreement surfaces. In English, for example, there is never agreement on the past participle[1], while in Italian[2], participles inflect like adjectives both when they are adjectival (7) and when they are verbal (8). The relevant morphology is in bold.

(7)　a.　Il　bicchiere sembra　rott-**o**
　　　　　the glass　M　seem.3SG broken-MSG
　　　　　'The glass seems broken'
　　b.　La　porta sembra　rott-**a**
　　　　　the door　seem.3SG broken-FSG
　　　　　'The door seems broken'

(8)　a.　Il bicchiere　è　　stat-o　　rott-**o**　　da Gianni
　　　　　the glass　　be.3SG been-MSG broken-MSG by John
　　　　　'The glass has been broken by John'
　　b.　La porta è　　stat-a　rott-**a**　　da Gianni
　　　　　the door be.3SG been-FSG broken-FSG by John
　　　　　'The door has been broken by John'

Arbëresh and Albanian only have past participial forms, similar to the *-ed* form in English, which has both verbal and adjectival properties[3]. As a verbal form, the participle is used in compound and in periphrastic tenses[4]. In compound tenses, the participle can combine with both the auxiliaries *kam* 'have' and *jam* 'be'. Arbëresh and Albanian use *kam* 'have' for transitives (9a), unergatives (9b), and unaccusatives (9c)[5]:

(9)　a.　Meri　　　ka　　**marrë** lul-e-t
　　　　　Mary.NOM have.3SG taken flower-PL-DEF.ACC[6]
　　　　　'Mary has taken the flowers'
　　b.　Meri　　　ka　　**qeshur**
　　　　　Mary.NOM have.3SG laughed'
　　　　　'Mary has laughed'
　　c.　Meri　　　ka　　**ardhur**
　　　　　Mary.NOM have.3SG arrived
　　　　　'Mary has arrived'

In Italian, the past participle, when combined with the auxiliary *avere* 'have', is used to form compound verbal tenses of transitives (10a) and unergatives (10b):

(10)　a.　Maria ha　　preso i　fiori
　　　　　Mary have.3SG taken the flowers
　　　　　'Mary has taken the flowers'
　　b.　Maria ha　　riso
　　　　　Mary have.3SG laughed'
　　　　　'Mary has laughed'

Unaccusative verbs, instead, select the auxiliary *essere* 'be' and require agreement with the subject on the past participle:

(11)　a.　Maria è　　arrivat-**a**
　　　　　Mary be.3SG arrived-FSG
　　　　　'Mary has arrived'
　　b.　*Maria è　　arrivat-**o**
　　　　　Mary be.3SG arrived-MSG
　　　　　'Mary has arrived'

In compound tenses, the Arbëresh/Albanian participle is uninflected, i.e., it agrees neither with the subject (9) nor with the object/theme (12), so it makes no distinction between masculine and feminine or singular and plural:

(12)  a.    Kam    **marrë** libr-i-n
             have.1SG taken  book-MSG.DEF-ACC
             'I took the book'

      b.    Kam    **marrë** çantën
             have.1SG taken  bag-FSG.DEF.ACC
             'I took the bag'

      c.    Kam    **marrë** libr-a-t
             have.1SG taken  book-MPL-DEF.ACC
             'I took the books'

      d.    Kam    **marrë** çant-a-t
             have.1SG taken  bag-FPL-DEF.ACC
             'I took the bags'

The participle does not agree even when the direct object is moved to the sentence's initial position and is resumed by a clitic:

(13)  a.    Libr-i-n        e        kam    **marrë**
             book-MSG.DEF-ACC  CL.ACC have.1SG taken
             'The book, I took it'

      b.    Çant-ën       e        kam    **marrë**
             bag-FSG.DEF.ACC  CL.ACC have.1SG taken
             'The bag, I took it'

Agreement is instead obligatory in Italian when the direct object is left-dislocated to the front of the sentence, and it is doubled by a clitic[7]:

(14)  a.    L'auto,    l'ho        lavat-**a**    /\*lavat-**o**
             the car.FSG CL+have.1SG washed-FSG /washed-MSG
             'The car, I washed it'

      b.    Il    vestito,   l'ho        lavat-**o**    /\*lavat-**a**
             the dress.MSG CL+have.1SG washed-MSG/washed-FSG
             'The dress, I washed it'

Nothing can be inserted between the auxiliary and the participle, neither the direct object (15a) nor adverbial material (15b), as shown by the ungrammaticality of the following examples:

(15)  a.    \*Kam    makin-ën    larë
             have.1SG car-FSG.DEF.ACC washed
             'I washed the car'

      b.    \*Kam    gjithmonë larë    makin-ën
             have.1SG always    washed car-FSG.DEF.ACC
             'I always washed the car'

A partially similar pattern is found in Italian, where the adjacency between the auxiliary and the participle cannot be interrupted, for example, by a noun phrase (16a) or by some adverbials like *oggi* 'today'(16b), whereas it is possible to insert adverbials like *sempre* 'always' or *spesso* 'often' (16c).

(16)  a.    \*Ho    l'auto  lavato
             have.1SG the car washed
             'I washed the car'

      b.    \*Ho    oggi   lavato  l'auto
             have.1SG today washed the car
             'I washed the car today'

      c.    Ho     sempre/spesso lavato l'auto
             have.1SG always/often  washed the car
             'I always/often washed the car'

In compound tenses, it is possible to move the direct object, but only in a position preceding the auxiliary. This movement reflects a difference in information structure. Preposing of the object corresponds to focalisation (17a) or left dislocation, with a co-

referential resumptive pronoun (17b), but without agreement on the participle. Also in these cases, the adjacency between the auxiliary and the participle is not interrupted.

(17) a.    MAKIN-ËN    kam    **larë**,    jo këmish-ën
         car-FSG.DEF.ACC have.1SG washed not shirt-FSG.DEF.ACC
         'I washed the car, not the shirt'
     b.    Makin-ën      e      kam    **larë**
         car-FSG.DEF.ACC CL.ACC have.1SG washed
         'I washed the car'

With the auxiliary *jam* 'be', the participle is used to form the compound tenses of non-active verbs, both in Arbëresh and Albanian. Example (18) is the non-active present perfect of the verb *laj* 'wash':

(18)    Meri      është    **larë**
         Mary.NOM be.3SG washed
         'Mary washed'

In standard Albanian, the non-active verbal form characterises reflexives (18), anticausatives (19a), middles (19b), and passives (19c)[8]:

(19) a.    Der-a        është hapur
         door-DEF.NOM be.3SG opened
         'The door opened'
     b.    Në atë   restorant është ngrënë mirë
         in   that restaurant be.3SG eaten   well
         'In that restaurant we ate well'
     c.    Libr-i        është lexuar nga Ben-i      /prej Ben-it[9]
         book-DEF.NOM be.3SG read   by Ben-NOM/by   Ben-ABL
         'The book is read by Ben'

The Arbëresh variety considered here shares with standard Albanian the syntactic structure of reflexives, unaccusatives, and middles, but not the structure of passives, which will be investigated in Section 7.

Coming back to the compound tenses formed with the auxiliary *jam* 'be', we can see in (20) that, in this case, too, the participle is uninflected, so it does not agree with the subject of the sentence:

(20) a.    Djal-i        është    **larë**
         boy-MSG.DEF.NOM be.3SG washed
         'The boy washed'
     b.    Vajz-a        është    **larë**
         girl-FSG.DEF.NOM be.3SG washed
         'The girl washed'
     c.    djem-të      janë    **larë**
         boy.MPL-DEF.NOM be.3PL washed
         'The boys washed'
     d.    Vajz-a-t      janë    **larë**
         girl-FPL-DEF.NOM be.3PL washed
         'The girls washed'

The paradigm is different from the one we find in Italian, where the presence of the auxiliary *essere* 'be' imposes the agreement on the past participle:

(21) a. Il ragazz-o si è lavat-**o**
the boy-MSG REFL be.3SG washed-MSG
'The boy washed'

b. La ragazz-a si è lavat-**a**
the girl-FSG REFL be.3SG washed-FSG
'The girl washed'

c. I ragazz-i si sono lavat-**i**
the boy-MPL REFL be.3PL washed-MPL
'The boys washed'

d. Le ragazz-e si sono lavat-**e**
the girl-FPL REFL be.3PL washed-FPL
'The girls washed'

Briefly, the verbal participle that occurs in Arbëresh/Albanian compound tenses never agrees with the subject or the object of the sentence.

## 3. Adjectival Participles

### 3.1. The Structure of Arbëresh/Albanian Adjectives

As we saw, in addition to having verbal properties, participles also have adjectival properties[10]. Before showing the contexts where they are adjectival, I will present the basic properties of Arbëresh and Albanian adjectives; then, I will consider the adjectival use of the participles.

Arbëresh and Albanian have two classes of adjectives: the class of adjectives which appear with a prefix, a free article, which agrees with the noun in gender (masculine or feminine), number (singular or plural), definiteness (definite or indefinite), and Case (nominative, genitive, dative, accusative, ablative)[11], and a second class, without an article, that only displays gender and number inflection. The first class is exemplified in (22)–(25). In (22), the pre-articulated adjective modifies a definite masculine singular noun which is, respectively, marked with a nominative, accusative, and dative Case[12]:

(22) a. Djal-**i** **i** **urtë** flet pak
boy-MSG.DEF.NOM MSG.NOM wise speak.3SG little
'The wise boy speaks little'

b. Takova djal-i-**n** **e** **urtë**[13]
meet.1SG.AOR boy-MSG.DEF-ACC ACC wise
'I met the wise boy'

c. I dashë libr-i-n djal-i-**t** **të** **urtë**
CL.DAT.SG give.1SG.AOR book-MSG.DEF-ACC boy-MSG.DEF-OBL MSG.OBL wise
'I gave the book to the wise boy'

In (23), the pre-articulated adjective modifies a definite feminine singular noun:

(23) a. Vajz-**a** **e** **urtë** flet pak
girl-FSG.DEF.NOM FSG.NOM wise speak.3SG little
'The wise girl speaks little'

b. Takova vajz-ë**n** **e** **urtë**[14]
meet.1SG.AOR girl-FSG.DEF.ACC ACC wise
'I met the wise girl'

c. I dashë libr-i-n vajz-ë**s** **së** **urtë**[15]
CL.DAT.SG give.1SG.AOR book-MSG.DEF-ACC girl-FSG.DEF.OBL FSG.OBL wise
'I gave the book to the wise girl'

In (24a), the pre-articulated adjective modifies a definite masculine plural noun, in the nominative and in the accusative Case. In (24b), the pre-articulated adjective modifies a definite masculine plural noun in the oblique Case:

(24) a. Djem-**të**    **e**    **urt**ë flasin    pak[16]
     boy.MPL-DEF.NOM PL.NOM wise speak.3PL little
     'The wise boys speak little'

     b. Takova    djem-**të**    **e**    **urtë**
     meet.1SG.AOR boy.MPL-DEF.ACC PL.ACC wise
     'I met the wise boys'

     c. U    dashë    libr-a-t    djem-**ve**    **të**    **urtë**
     CL.DAT.PL give.1SG.AOR book-MPL-DEF.ACC boy.MPL-DEF.OBL PL.OBL wise
     'I gave the books to the wise boys'

In (25a), the pre-articulated adjective modifies a definite feminine plural noun, in the nominative and in the accusative Case. In (25b), the pre-articulated adjective modifies a definite feminine plural noun in the oblique Case:

(25) a. Vajz-a-**t**    **e**    **urt-a** flasin    pak
     girl-FPL-DEF.NOM   FPL.NOM wise-FPL speak.3PL little
     'The wise girls speak little'

     b. Takova    vajz-a-**t**    **e**    **urt-a**
     meet.1SG.AOR girl-FPL-DEF.ACC FPL.ACC wise-FPL
     'I met the wise girls'

     c. U    dashë    libr-a-t    vajz-a-**ve**    **të**    **urt-a**
     CL.DAT.PL give.1SG.AOR book-MPL-DEF.ACC girl-FPL-DEF.OBL FPL.OBL wise-FPL
     'I gave the books to the wise girls'

The adjectives without article are exemplified in (26)–(29).

(26) a. Djal-**i**    **guximtar** lufton    pa    frikë
     boy-MSG.DEF.NOM brave.MSG fight.3SG without fear
     'The brave boy fights without fear'

     b. Presidenti    takoi    djalin    **guximtar**
     president-MSG.DEF.NOM meet.3SG.AOR boy-MSG.DEF-ACC brave.MSG
     'The president met the brave boy'

     c. I    dha    medhalj-en    djal-i-**t**    **guximtar**
     CL.DAT.SG give.3SG.AOR medal-FSG.DEF.ACC boy-MSG.DEF-OBL brave.MSG
     'He gave the medal to the brave boy'

(27) a. Vajz-**a**    **guximtar-e** lufton    pa    frikë
     girl-FSG.DEF.NOM brave-FSG fight.3SG without fear
     'The brave girl fights without fear'

     b. Presidenti    takoi    vajzën    **guximtar-e**
     president-MSG.DEF.NOM meet.3SG.AOR girl-FSG.DEF.ACC brave-FSG
     'The president met the brave girl'

     c. I    dha    medhalj-en    vajz-ë**s**    **guximtar-e**
     CL.DAT.SG give.3SG.AOR medal-FSG.DEF.ACC girl-FSG.DEF.OBL brave-FSG
     'He gave the medal to the brave girl'

(28) a. Djemtë    **guximtar-ë** luftojnë pa frikë
     boy.MPL-DEF.NOM brave-MPL   fight.3PL without fear
     'The brave boys fight without fear'

     b. President-i    takoi    djem-**të**    **guximtar-ë**
     president-MSG.DEF.NOM meet.3SG.AOR boy.MPL-DEF.ACC brave.MPL
     'The president met the brave boys'

     c. U    dha    medhalj-en    djem-**ve**    **guximtar-ë**
     CL.DAT.PL give.3SG.AOR medal-FSG.DEF.ACC boy.MPL-DEF.OBL brave.MPL
     'He gave the medal to the brave boys'

(29)  a.  Vajz-a-**t**       **guximtar-e** luftojnë  pa     frikë
         girl-FPL-DEF.NOM brave-FPL  fight.3PL without fear
         'The brave girls fight without fear'

      b.  President-i        takoi        vajz-a-**t**       **guximtar-e**
         president-MSG.DEF.NOM meet.3SG.AOR girl-FPL-DEF.ACC  brave-FPL
         'The president met the brave girls'

      c.  U        dha        medhalj-en     vajz-a-**ve**    **guximtar-e**
         CL.DAT.PL give.3SG.AOR medal-FSG.DEF.ACC girl-FPL-DEF.OBL brave-FPL
         'He gave the medal to the brave girls'

As we can see, the adjectives without an article are only sensitive to gender and number. Their forms are the same for all the different Cases. They remain unchanged also in relation to definiteness.

Arbëresh and Albanian adjectives occur in a post-nominal position, both with definite and indefinite nouns. Example (30) shows the case of a pre-articulated adjective. Example (31) contains an adjective without an article:

(30)  a.  Djal-i       /një djalë i  urtë
         boy-MSG.DEF/ a   boy  MSG wise
         'The/a wise boy'

      b.  *I urtë djali/*Një i urtë djalë

(31)  a.  Djal-i       /një djalë idiot
         boy-MSG.DEF /a   boy  idiot
         'The/an idiot boy'

      b.  *Idiot djali/*Një idiot djalë

In Arbëresh and Albanian, both attributive and predicative adjectives agree in gender, number, and Case with the noun they modify[17]. The examples in (32) contain attributive adjectives (masculine and feminine) in the nominative Case. The examples in (33) exemplify the accusative Case. The examples in (34) correspond to the oblique Case.

(32)  a.  Djal-i        **i**      **urtë** shkon në shtëpi
         boy-MSG.DEF.NOM  MSG.NOM  wise  go.3SG in house
         'The wise boy goes home'

      b.  Vajz-a       **e**      **urtë** shkon në shtëpi
         girl-FSG.DEF.NOM  FSG.NOM  wise  go.3SG in house
         'The wise girl goes home'

(33)  a.  Takova     djal-i-n       **e**      **urtë**
         meet.1SG.AOR boy-MSG.DEF-ACC MSG.ACC wise
         'I met the wise boy'

      b.  Takova     vajz-ën     **e**      **urtë**
         meet.1SG.AOR  girl-FSG.DEF.ACC FSG.ACC wise
         'I met the wise girl'

(34)  a.  I        dashë     libr-i-n       djal-i-t      **të**     **urtë**
         him.CL.DAT give.1SG.AOR book-MSG.DEF-ACC boy-MSG.DEF-OBL MSG.OBL wise
         'I gave the book to the wise boy'

      b.  I        dashë     libr-i-n       vajz-ës     **së**     **urtë**
         her.CL.DAT give.1SG.AOR book.MSG.DEF-ACC  girl-FSG.DEF.OBL FSG.OBL wise
         'I gave the book to the wise girl'

(35)  a.  Djal-i        është  **i**     **urtë**
         boy-MSG.DEF.NOM be.3SG  MSG.NOM wise
         'The boy is wise'

      b.  Vajz-a       është  **e**     **urtë**
         girl-FSG.DEF.NOM  be.3SG FSG.NOM wise
         'The girl is wise'

*3.2. Adjectival Participles*

In Albanian and Arbëresh, all the adjectives derived by a participial verb take the linking article:

(36)  a.    i banuar      'inhabited'
       b.    i botuar       'published'
       c.    i ftuar         'invited'
       d.    i harruar     'forgotten'
       e.    i larë         'washed'
       f.     i martuar     'married'
       g.    i paguar     'paid'
       h.    i thyer       'broken'

The article is obligatory. Its absence gives rise to ungrammaticality:

(37)  *Djal-i        martuar/*larë/*plagosur/*harruar
       boy-MSG.DEF married/washed/blessed/forgotten
       'The married/washed/blessed/forgotten boy'

These adjectives of participial origin always agree with the noun they modify in gender, number, Case, and definiteness.

(38)  a.    Djal-i           **i**       **martuar** punon shumë
           boy-MSG.DEF.NOM  MSG.NOM  married work-3SG much
           'The married boy works a lot'
       b.    Takova        djalin               **e martuar**
           meet.1SG.AOR boy-MSG.DEF-ACC MSG.ACC  married
           'I met the married boy'
       c.    I           dashë    libr-i-n        djal-i-t       **të**      **martuar**
           CL.DAT.SG. give.1SG book.DEF-ACC boy-MSG.DEF-OBL MSG.OBL married
           'I gave the book to the married boy'

(39)  a.    Vajz-a          **e**         martuar punon    shumë
           girl-FSG.DEF.NOM  FSG.NOM  married work-3SG much
           'The married girl works a lot'
       b.    Takova        vajz-ën        **e**       **martuar**
           meet.1SG.AOR girl-FSG.DEF.ACC FSG.ACC married
           'I met the married girl'
       c.    I           dashë    libr-i-n        vajz-ës       **së**      **martuar**
           CL.DAT.SG. give.1SG book-MSG.DEF-ACC girl-FSG.DEF.OBL FSG.OBL married
           'I gave the book to the married girl'

The pre-articulated adjectives derived by a participial verb can be prefixed with the negation *pa* ('un'-), which cannot be attached to the other adjectives[18]. Compare the adjectives in (40) with those in (41):

(40)  a.    (i) pabanuar     'uninhabited'
       b.    (i) pabotuar     'unpublished'
       c.    (i) paftuar       'uninvited'
       d.    (i) paharruar    'unforgotten'
       e.    (i) palarë       'unwashed'
       f.     (i) pamartuar    'unmarried'
       g.    (i) papaguar     'unpaid
       h.    (i) pathyer      'unbroken'

(41)  a.    *(i) pabardh     'not white'
       b.    *(i) paëmbël     'not sweet'
       c.    *(i) pamadh     'not big'
       d.    *(i) pashkurtër  'not short'
       e.    *(i) pavogël    'not small'

Participial adjectives can appear in attributive and predicative positions:

(42)  a.  Libr-i          **i**      **botuar**    në Tiranë
          book-MSG.DEF.NOM MSG.NOM published in Tirana
          'The book published in Tirana'

      b.  Libr-i          është **i**    **botuar**
          book-MSG.DEF.NOM be.3SG MSG.NOM published
          'The book is published'

Italian, too, has adjectives derived from participles. They must agree with the noun they modify according to gender (masculine or feminine) and number (singular or plural), both in attributive (43) and predicative (44) positions.

(43)  a.  Il ragazz-o sposat-**o**    lavora   a Tirana
          the boy-MSG  married-MSG  work.3SG in Tirana
          'The married boy works in Tirana'

      b.  La ragazz-a sposat-**a**   lavora   a Tirana
          the girl-FSG married-FSG work.3SG  in Tirana
          'The married girl works in Tirana'

      c.  I ragazz-i  sposat-**i**    lavorano a Tirana
          the boy-MPL  married-MPL work.3PL in Tirana
          'The married boys work in Tirana'

      d.  Le ragazz-e sposat-**e**    lavorano a Tirana
          the girl-FPL   married-FPL work.3PL in Tirana
          'The married girls work in Tirana'

(44)  a.  Il ragazz-o  è     sposat-**o**
          the boy-MSG  be.3SG married-MSG
          'The boy is married'

      b.  La ragazz-a  è     sposat-**a**
          the girl-FSG  be.3SG married-FSG
          'The girl is married'

      c.  I ragazz-i  sono   sposat-**i**
          the boy-MPL  be.3PL  married-MPL
          'The boys are married'

      d.  Le ragazz-e  sono  sposat-**e**
          the girl-FPL   be.3PL married-FPL
          'The girls are married'

## 4. Eventive vs. Stative Participles in Standard Albanian

As we saw, Albanian/Arbëresh participles can be both verbal and adjectival. Verbal participles never show agreement (cfr. (12)), while adjectival participles obligatorily agree with the noun they modify (cfr. (38) and (39)).

At first sight, this formal distinction of verbal participles from adjectival participles seems to correlate with the aspectual properties of the construction: (45a) and (46a), which contain a verbal participle, correspond to eventive structures, while (45b) and (46b), containing an adjectival form, are statives, since they describe the state resulting from the event expressed by the verb[19].

(45)  a.  Der-a          është **hapur**
          door-FSG.DEF.NOM be.3SG open
          'The door opened'

      b.  Der-a          është **e**   **hapur**
          door-FSG.DEF.NOM be.3SG FSG.NOM open
          'The door is open'

(46)  a.  Fush-a-t       janë  **mbuluar** me ujë
          field-FPL-DEF.NOM be.3PL covered with water
          'The fields covered with water'

      b.  Fush-a-t       janë **të**   **mbuluar-a** me ujë
          field-FPL-DEF.NOM be.3PL PL.NOM covered-FPL with water
          'The fields are covered with water'

A series of tests has been used in order to show the distinction between eventive and stative interpretation. I will apply some of these tests to standard Albanian to show that Albanian, too, supports this traditional classification. Then, I will come back to Arbëresh.

The first test I consider for separating stative participles from eventive participles is their distribution after some verbs of creation, such as *build*, *make*, and *create* (Embick 2004). Only stative inflected participles can appear in the complement of these verbs; eventive uninflected participles cannot.

(47)  a.  Der-a        ishte      ndërtuar **e**    **mbyllur**
           door-FSG.DEF.NOM be.3SG.IMPF built    FSG.NOM closed
           'The door was built closed'
      b.  *Dera ishte ndërtuar **mbyllur**

Statives are compatible with the adverbial 'still', while eventives are not. In (48), *akoma/ende* 'still' forces a stative interpretation of the participle.

(48)  a.  Meri  është akoma/ende **e**    **martuar**
           Mary be.3sg    still    FSG.NOM married
           'Mary is still married'
      b.  *Meri është akoma/ende **martuar**

Stative participles can appear in the complement of a raising verb like *seem* (Wasow 1977); eventives cannot.

(49)  a.  Der-a          duket    **e mbyllur**
           door-FSG.DEF.NOM seem.3SG FSG  closed
           'The door seems closed'
      b.  Vazo-ja       duket    **e thyer**
           vase-FSG.DEF.NOM seem.3SG  FSG broken
           'The vase seems broken'
      c.  *Dera duket **mbyllur**
      d.  *Vazoja duket **thyer**

Stativity requires an adjectival participle that agrees with the noun, both in attributive (50) and predicative interpretations (45b):

(50)  a.  Nga der-a          **e**    **hapur** hyjnë    dy vajza
           from door-FSG.DEF.NOM FSG.NOM open   enter.3PL two girls
           'Two girls enter through the open door'
      b.  *Nga dera **hapur** hyjnë dy vajza

Another indication of stativity comes from *haben*-passives, i.e., constructions whose structure is *have DP V-ed*. They are familiar in English (51a) as well as in Italian (51b):

(51)  a.  Ben has his leg broken
      b.  Ben ha        la gamba rotta
           Ben have.3SG the leg   broken
           'Ben has his leg broken'

In these constructions, which are characterised by a possessive-like relationship of the subject with the object, the participle expresses the state of the object.

In Albanian *haben*-passives, agreement on the participle is obligatory as we expect it to be in statives:

(52)  a.  Beni ka    këmb-ën        **e**     **thyer**
           Ben have.3SG leg-FSG.DEF.ACC FSG.ACC broken
           'Ben has his leg broken'
      b.  *Beni ka këmbën **thyer**

Eventive participles, instead, display other characteristics. They can be modified by some temporal adverbials, while statives cannot[20].

(53) a. Meri është **martuar** dje
Mary be.3SG married yesterday
'Mary (got) married yesterday'
   b. *Meri është **e martuar** dje
Mary be.3SG FSG married yesterday

Eventive participles can be modified by a PP-agent, while stative participles are incompatible with it:

(54) a. Hajdut-i       është **arrestuar** nga polic-i
thief-MSG.DEF.NOM be.3SG arrested by policeman-MSG.DEF
'The thief was arrested by the police'
   b. *Hajduti është **i arrestuar** nga polici

Another test that has been used in the literature in order to show the difference between statives and eventives is the possibility/impossibility of allowing prefixation with un-. Un-prefixation is generally admitted with eventives but it is very restricted (Wasow 1977; Embick 2004) or impossible (Kratzer 1994) with statives.

(55) a. The island was uninhabited by humans
   b. The unopened door
   c. *The unopen door

In Albanian, un-prefixation is not a good diagnostic since it is compatible with both eventives (56) and statives (57).

(56) a. Ajo erdhi       **e paftuar** nga askush
she come.3SG.AOR FSG uninvited by nobody
'She came uninvited by anyone'
   b. Ishull-i       është **i pabanuar** nga njerëzit
island-MSG.DEF.NOM be.3SG MSG uninhabited by people
'The island is uninhabited by humans'

(57) a. Meri është **e pamartuar**
Mary be.3SG FSG unmarried
'Mary is unmarried'
   b. Fustan-i       është **i palarë**
dress-MSG.DEF.NOM be.3SG MSG unwashed
'The dress is unwashed'

Another useful test to show the difference between statives and eventives is provided by the behaviour of the verb *remain*. Embick (2004) has observed that the verb *remain* only takes stative complements and, as such, it cannot license agentive by-phrases.

(58) a. *The door remained opened by John
   b. The door remained opened

The corresponding Albanian verb *mbes*, like the English verb *remain*, does not support agentive by-phrases.

(59) a. *Der-a     mbeti       **e hapur** nga Beni
door-FSG.DEF remain.3SG.AOR FSG opened by Ben
*The door remained opened by Ben
   b. Dera mbeti **e hapur**

All these tests seem to prove that Albanian, too, has the traditional dichotomy of stative vs. eventive participles. These tests also show that stative contexts require inflected participles, while eventive sentences require verbal participles.

At first sight, it seems that agreement is obligatory when the construction is stative, while uninflected participles occur in eventive constructions. However, a closer look into the properties of Albanian shows that this conclusion is inappropriate because, in this language, stativity does not always imply adjectivehood. In Albanian, even sentences whose stativity is obvious can be realised with a verbal participle that occurs in an uninflected form. This

is shown precisely by the behaviour of the stative verb *mbes* 'remain'. As we can see in the examples in (60), in Albanian, agreement on the participle is not obligatory. All the sentences are fully grammatical.

(60) a. Der-a     mbeti     **hapur /e**    **hapur**
      door-FSG.DEF remain.3SG.AOR opened/ FSG opened
      'The door remained opened'

     b. Vajz-a     mbeti     **ulur /e ulur**    mbi një karrige
      girl-FSG.DEF remain.3SG.AOR seated/FSG seated on   a chair
      'The girl remained seated on a chair'

What Albanian shows is that the participle occurring in statives can be both verbal and adjectival. Other examples of stative sentences are in (61)–(63). As we can see, they can contain both a verbal uninflected participle or an adjectival inflected participle. There is no semantic difference between the two sentences of each pair.

(61) a. Shikonte    një vajzë **të**     **veshur** me fustan     të      kuq
      look.3SG.IMPF a   girl.FSG FSG.INDEF dressed with dress.FSG FSG.INDEF red
      'He was looking at a girl wearing a red dress'

     b. Shikonte një vajzë **veshur** me fustan të kuq

(62) a. Ushtar-i       mbeti      me duar-t      **e ngritur-a**
      soldier-MSG.DEF.NOM remain.3SG.AOR with hand.FPL-DEF.ACC FPL raised-FPL
      'The soldier remained with his hands up'

     b. Ushtari mbeti me duart **ngritur**

(63) a. Erdhi     një djalë **i lyer**      me baltë
      came.3SG.AOR a   boy.MSG MSG smeared with mud
      'A boy smeared with mud came'

     b. Erdhi një djalë **lyer** me baltë

In these sentences, which describe the state of the DPs *vajzë* 'girl', *duart* 'the hands', and *djalë* 'boy', the participle can appear with or without agreement.

Summing up, in standard Albanian, participles never inflect when they are verbal, i.e., when they occur in eventive structures (in transitives, unaccusatives, middles, and passives). Conversely, in stative sentences, participles can be both inflected and uninflected.

## 5. Eventive and Stative Participles in the Arbëresh Variety of S. Nicola dell'Alto in Comparison with Standard Albanian and Italian

At first sight, Arbëresh seems to be similar to Albanian since it requires an uninflected verbal participle in eventive structures (a-examples) and an inflected adjectival participle in stative sentences (b-examples):

(64) a. Maria osht **martuer**
      Mary be.3SG married
      'Mary (got) married'

     b. Maria osht **e martuer**
      Mary be.3SG  FSG married
      'Mary is married'

(65) a. Der-a       osht **hapur**
      door-FSG.DEF be.3SG open
      'The door opened'

     b. Der-a       osht **e hapur**
      door-FSG.DEF be.3SG FSG open
      'The door is open'

(66)  a.   Tac-a        osht  **çar**
          cup-FSG.DEF be.3SG broken
          'The cup broke'
      b.   Tac-a        osht   **e   çar**
          cup-FSG.DEF be.3SG FSG broken
          'The cup is broken'

Evidence for this distinction comes from modification. Arbëresh statives, built up by means of an inflected participle, allow for the presence of the adverbials *ankora* 'still' and *sempre* 'always' (67)[21], while eventives, which contain a verbal uninflected participle, do not support them (68):

(67)  a.   Maria osht ankora/sempre **e martuer**
          Mary be.3SG still/always    FSG married
          'Mary is still/always married'
      b.   Der-a          osht ankora/sempre **e  hapur**
          door-FSG.DEF.NOM be.3SG still/always    FSG open
          'The door is still/always opened'
      c.   Tac-a          osht ankora/sempre **e  çar**
          cup-FSG.DEF.NOM be.3SG still/always    FSG broken
          'The cup is still/always broken'

(68)  a.   *Maria osht ankora/sempre **martuer**
      b.   *Dera osht ankora/sempre **hapur**
      c.   *Taca osht ankora/sempre **çar**

Stative inflected participles can appear in the complement of the raising verb *duket* 'seem'; eventives cannot.

(69)  a.   Maria duket   **e martuer**
          Mary seem.3SG FSG married
          'Mary seems married'
      b.   Der-a      duket   **e hapur**
          door-FSG.DEF seem.3SG FSG open
          'The door seems open'
      c.   Tac-a      duket **e çar**
          cup-FSG.DEF seem.3SG FSG broken
          'The cup seems broken'

(70)  a.   *Maria duket **martuer**
      b.   *Dera duket **hapur**
      c.   *Taca duket **çar**

Statives do not support agentive PPs, but they can contain instrumental PPs:

(71)  a.   *Ghiter-a     osht e shkruer ka Maria
          letter-FSG.DEF be.3SG FSG written by Mary
          'The letter is written by Mary'
      b.   Ghiter-a     osht e  shkruer me laps-i-n
          letter-FSG.DEF be.3SG FSG written with pencil-MSG.DEF-ACC
          'The letter is written with the pencil'

Eventive participles, on the other hand, can be modified by temporal adverbials like *yesterday* or *on Sunday*, while statives cannot.

(72)  a.   Maria osht   **martuer** dje/të djelin
          Mary  be.3SG married yesterday/on Sunday
          'Mary (got) married yesterday/on Sunday'
      b.   *Maria osht **e martuar** dje/të djelin

All these examples show that Arbëresh statives require the participle to be obligatorily inflected, in contrast with eventives that contain verbal participles.

Up to now, the Arbëresh dialect considered here seems to behave like standard Albanian (cfr. (45)). However, it differs from Albanian in two points: (1) in Arbëresh,

the sentences whose stativity is obvious can only be realised with an adjectival participle that occurs in an inflected form. (2) In Arbëresh, eventive participles are verbal in reflexives, anticausatives, and middles but not in passives, which require inflected adjectival participles. In both these two points, Arbëresh does not match standard Albanian but the dominant language, Italian. The kind of variation present in the Arbëresh dialect is a clear case of contact-induced change. The Italian grammar is considered to be mainly involved in the change in the Arbëresh participial constructions.

As for the first point, if we compare Arbëresh with its genetically related language, Albanian, we can see that the Arbëresh structures having a stative interpretation differ from their counterparts in the Albanian examples in (61)–(63), since stativity, in Arbëresh, requires adjectival participles. So, in this variety, the stative examples with uninflected participles are all ungrammatical.

(73)  a.  *Vrehj         një vajzë **veshur** me vest        të         kuq-e
          look.3SG.IMPF a   girl.FSG dressed with dress.FSG FSG.INDEF red-FSG
          'He was looking at a girl wearing a red dress'
      b.  *Surdat-i              rrihj              me dor-a-t            **ngretur**
          soldier-MSG.DEF.NOM remain.3SG.IMPF with hand-FPL-DEF.ACC raised
          'The soldier remained with his hands up'
      c.  *Ka        ardhur një djalë **lyer**    me botë
          have.3SG came    a boy.MSG  smeared with mud
          'A boy smeared with mud came'

These examples become grammatical if the uninflected verbal participle is replaced with the adjectival one, i.e., a pre-articulated adjective that agrees with the noun it accompanies.

(74)  a.  Vrehj          një vajzë **e veshur** me vest        të         kuq-e
          look.3SG.IMPF a girl.FSG FSG dressed with dress.FSG FSG.INDEF red-FSG
          'He was looking at a girl wearing a red dress'
      b.  Surdat-i              rrihj            me dor-a-t            **e ngretur-a**
          soldier-MSG.DEF.NOM remain3SG.IMPF with hand-FPL-DEF.ACC FPL raised-FPL
          'The soldier remained with his hands up'
      c.  Ka        ardhur një djalë **i lyer**    me botë
          have.3SG came    a boy.MSG  MSG smeared with mud
          'A boy smeared with mud came'

So, while Albanian can have an uninflected participle in stative contexts, this Arbëresh variety cannot. It must always have inflected participles.

The same phenomenon is present in Italian, where stative constructions display obligatory agreement on the lexical participle (75), while the uninflected versions are ungrammatical (76).

(75)  a.  La port-a è       apert-**a**
          the door-FSG be.3SG open-FSG
          'The door is open'
      b.  Le port-e sono    apert-**e**
          the door-FPL be.3PL open-FPL
          'The doors are open'
      c.  Il cancell-o è       apert-**o**
          the gate-MSG be.3SG open-MSG
          'The gate is open'
      d.  I cancell-i sono    apert-**i**
          the gate-MPL be.3PL open-MPL
          'The gates are open'

(76)  a.  *La porta è aper**to**
      b.  *Le porte sono aper**to**
      c.  *I cancelli sono aper**to**

That in Arbëresh stativity implies adjectivehood is further shown by the behaviour of a class of adjectival participles in -*m*, a suffix that attaches to a Romance/Italian base. These participles are pre-articulated. Some examples are in (77):

(77)  a.  i distruxhir**m** < distruxhiri < (It. *distruggere* 'to destroy')      'destroyed'
     b.  i ferir**m** < feriri < (It. *ferire* 'to hurt')      'hurt/injured'
     c.  i pitar**m** < pitari (Romance dialects *pittare* 'to paint')      'painted'
     d.  i dobar**m** < dobari (Romance dialects *dobbare* 'to repair')      'repaired'
     e.  i avizar**m** < avizari (It. avvisare 'to notify/advise')      'advised'
     f.  i fisar**m** < fisari (It. *fissare* 'to attach')      'attached'
     g.  i firmar**m** < firmari (It. *firmare* 'to sign')      'signed'
     h.  i shukar**m** < shukari (It. *asciugare* 'to dry')      'dried'
     i.  i gonfiar**m** < gonfiari (It. *gonfiare* 'to inflate')      'inflated'
     j.  i rovinar**m** < rovinari (It. *rovinare* 'to ruin')      'ruined'

Their characteristic is that they can only be used as adjectives in stative contexts:

(78)  a.  Shpi-a      osht   e   distruxhirm-e
         house-FSG.DEF be.3SG FSG destroyed-FSG
         'The house is destroyed'
     b.  Kuadr-i        osht   i   fisarm      te muri
         painting-MSG.DEF be.3SG MSG fixed.MSG at wall.DEF
         'The painting is fixed to the wall'
     c.  Ghiter-a     osht   e   firmarm-e
         letter-FSG.DEF be.3SG FSG signed-FSG
         'The letter is signed'

Their use in verbal contexts, such as the active perfect (79), or in reflexive/unaccusative sentences, is ungrammatical (80).

(79)  a.  *Ai ka       distruxhir**m** shpi-në
         he have.3SG destroyed  house-ACC.DEF
         'He has destroyed the house'
     b.  *Ai ka       fisar**m** kuadr-in      te muri
         he have.3SG fixed painting-ACC.DEF  at wall.DEF
         'He has fixed the painting to the wall'
     c.  *Ai ka       firmar**m** ghiter-in
         he have.3SG signed  letter-ACC.DEF
         'He has signed the letter'

(80)  a.  *Djal-i      osht ferir**m**
         boy-MSG.DEF be.3SG injured.MSG
         'The boy injured'
     b.  *Shpi-a     osht   distruxhirm-**e**
         house-FSG.DEF be.3SG  destroyed-FSG
         'The house destroyed'
     c.  *Vest-a-t     janë shukarm-**e**
         clothes-FPL-DEF be.3PL dried-FPL
         'The clothes dried'

In verbal contexts, only a –(*u*)*r* participle is possible, on the same verbal base:

(81)  a.  Ai ka       distruxhir**tur** shpi-në
         he have.3SG destroyed   house-ACC.DEF
         'He has destroyed the house'
     b.  Ai ka       fisar**tur** kuadr-in        te mur-i
         he have.3SG fixed   painting-ACC.DEF at wall-DEF
         'He has fixed the painting to the wall'
     c.  Ai ka        firmar**tur** ghiter-in
         he have.3SG signed  letter-ACC.DEF
         'He has signed the letter'

(82)  a.    Djal-i        osht  ferir**tur**
                boy-NOM.DEF be.3SG injured
                'The boy injured'
      b.    Shpi-a       osht   distruxhir**tur**
                house-NOM.DEF be.3SG destroyed
                'The house destroyed'
      c.    Vest-a-t      janë shukar**tur**
                clothes-PL-NOM.DEF be.3PL dried
                'The clothes dried'

The forms in *-m* do not support un-prefixation. This seems to confirm the general pattern that *un*-prefixation is not productive with statives.

(83)  a.    *Kuadr-i       osht  (i)  pafisar**m**
                painting-MSG.DEF be.3SG (MSG) unfixed.MSG
                'The painting is not fixed'
      b.    *Shpi-a      osht  (e)  papitarm-**e**
                house-FSG.DEF be.3SG (FSG) unpainted-FSG
                'The house is not painted'

Un-prefixation is instead possible when the −(*u*)*r* participle is used:

(84)  a.    Der-a        osht   padobar**tur**
                door-NOM.DEF be.3SG  unrepaired
                'The door is not repaired'
      b.    Shpi-a       osht   papitar**tur**
                house-NOM.DEF be.3SG  unpainted
                'The house is not painted'

The adjectival participles in *-m* show the same syntactic behaviour of those inherited from Albanian, when used in stative contexts: they can support the adverbial modifiers *ankora* 'still' o *sempre* 'always' (85a); they support instrumental PPs (85b); they can appear in the complement of the raising verb *duket* 'seem' (85c); they do not support an agentive PP (85d); and they do not support temporal adverbials like *yesterday* or *on Sunday* (85e).

(85)  a.    Der-a       osht ankora/sempre e rovinarm-e
                door-FSG.DEF be.3SG still/always    FSG ruined-FSG
                'The door is still/always ruined'
      b.    Ghiter-a     osht  e firmarm-e me laps-in
                letter-FSG.DEF be.3SG  FSG signed-FSG with pencil-ACC.DEF
                'The letter is signed with the pencil'
      c.    Der-a       duket  e rovinarm-e
                door-FSG.DEF seem.3SG FSG ruined-FSG
                'The door seems ruined'
      d.    *Der-a       osht  e  rovinarm-e ka Maria
                door-FSG.DEF be.3SG FSG ruined-FSG by Mary
                'The door is ruined by Mary'
      e.    *Ghiter-a    osht  e  firmarm-e të djelin
                letter-FSG.DEF be.3SG FSG signed-FSG on Monday
                'The letter is signed on Monday'

This class of adjectives seems to confirm that Arbëresh statives obligatorily require inflected participles.

The second point of difference between Arbëresh and Albanian is related to eventive participles of passive sentences, which will be presented in Section 7.

## 6. The Structure of Statives

As we saw, one of the tests traditionally used in order to distinguish statives from eventives, i.e., adjectival participles from verbal participles, is their behaviour with respect to certain types of modifiers. Statives are not compatible with modifiers like PP-agents or certain temporal adverbials, whereas eventives are compatible with both.

This difference seems to derive from the different basic structures of the two constructions and from their derivation: statives contain adjectival participles that are formed in the lexicon. They have a reduced structure or possibly no event structure, like adjectives (Wasow 1977; Rapp 1996; Kratzer 2000; Embick 2004). Statives behave like simple copular sentences that do not project an event. In statives, there is no verbal structure, therefore they do not allow the possibility to attach an agentive PP. Eventive participles, instead, are compatible with agentive PPs and this provides evidence for the presence of a verbal structure including a VoiceP projection or an AspP projection. Eventives such as *The glass was broken by John* have a structure containing an agent (*John*) and a theme (*glass*), similar to that of an active verb. Briefly, one of the ideas widely assumed in the literature is that adjectival participles are derived from the lexicon, while verbal participles are built in the syntax.

Kratzer (1994; 2000, p. 392), using data from German, argues that adjectival participles do not form a uniform class. Some of them involve a certain amount of verbal structure, as is shown by the fact that they can be modified by manner adverbials, while simple adjectives cannot.

(86)  a.  Die Haare waren immer noch schlampig gekämmt
          the hairs  were  still      sloppily  combed
          'The hair  was  still sloppily combed'
      b.  *Die Haare waren schlampig fettig
          'The hair was sloppily greasy'

For adjectival participles that are compatible with adverbial modification, she proposes a structure where a VP is embedded under an AP projection. Adverbials are adjoined to a VP. Since the adjectival head applies to the entire verb phrase, Kratzer refers to them as *Phrasal Adjectival Participles*. The structure she proposes, cited by Anagnostopoulou (2003, p. 6), is the following:

(87)

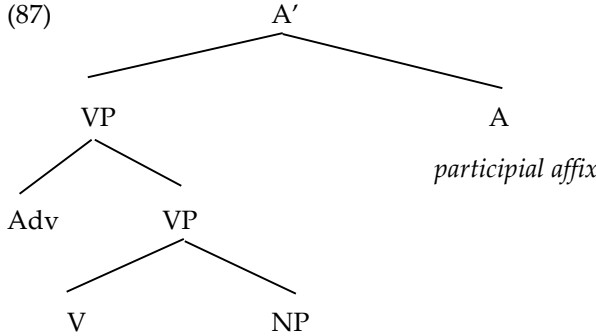

On the other hand, Kratzer (1994), cited by Anagnostopoulou (2003, p. 6), also notes that negated participles cannot be modified by adverbials:

(88)  *Das  Haar war hässlich ungekämmt
      '*The hair  was  ugly  uncombed'

These adjectival participles are *Lexical Adjectival Participles*, lacking the VP projection where adverbs should attach. They have the following reduced structure, reported by Anagnostopoulou (2003, p. 6):

(89)

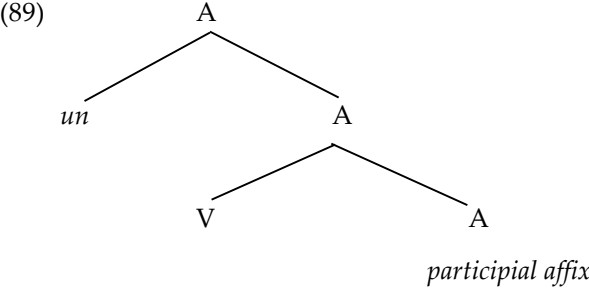

*participial affix*

Thus, adjectival participles can be both phrasal or lexical.

Verbal participles, instead, are compatible with agentive *by*-phrases. The presence of a PP-agent indicates that these sentences have a different structure, involving a VoiceP where the external argument is merged (Anagnostopoulou 2003, p. 7):

(90)

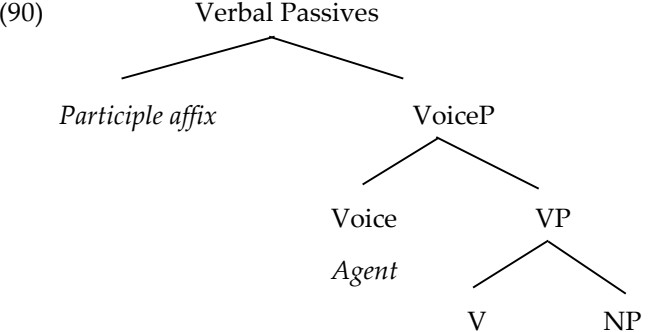

The idea that participles do not form a uniform class is also endorsed by Anagnostopoulou (2003), Alexiadou and Anagnostopoulou (2008), and Alexiadou et al. (2014) on the basis of some Greek data. These authors show that Greek has two morphologically distinct classes of adjectival participles: *-menos* and *-tos* participles. *-Menos* participles support both adverbial modifiers and agentive PPs because they have a phrasal structure. Adverbials and PPs are contained in the VP projection embedded under the participial affix *-menos*. Instead, *-tos* participles do not license adverbials and agentive PPs because they have the structure of lexical adjectival participles. The affix *-tos* attaches to V rather than to VP; therefore, there is no room available for the agentive PP. Furthermore, these authors, using a series of tests, and following Kratzer (2000), divide *-menos* participles in two subclasses: target and resultant state participles[22]. So, they identify three different types of adjectival participles: lexical *-tos* participles, without verbal/eventive structure; *-menos* target state participles; and *-menos* resultant state participles. All three types contain a stativizer head, but they differ in the position at which this head is attached. *-Tos* participles involve root attachment of the stativizer head (Alexiadou and Anagnostopoulou 2008, p. 38):

(91)

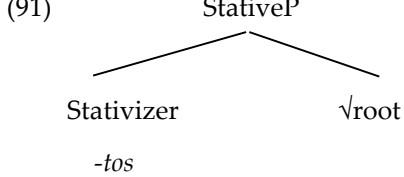

Target state *-menos* participles involve a vP projection embedded under the stativizer head. The v head is necessary to license the manner adverbs that can modify such constructions (Alexiadou and Anagnostopoulou 2008, p. 38):

(92)

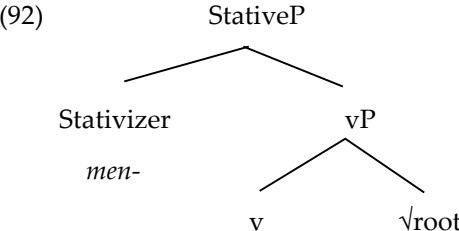

Resultant state *-menos* participles involve a VoiceP projection where agentive PPs can be hosted (Alexiadou and Anagnostopoulou 2008, p. 39):

(93)

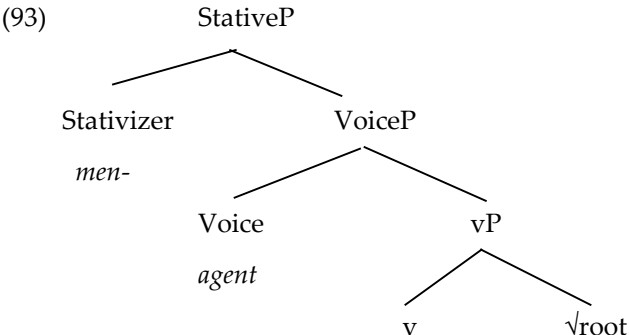

Verbal participles, on the other hand, differ from stative adjectival participles with respect to eventiveness (Anagnostopoulou 2003).

Albanian and Arbëresh participles have been studied by Manzini and Savoia (2018, p. 290), who assume that participles have a verb root followed by a thematic vowel in addition to an element that is the lexicalisation of an Asp category. Then, a participial form like that in (94a) has the structure in (94b):

(94) a.  Veshur

'Dressed'

b.

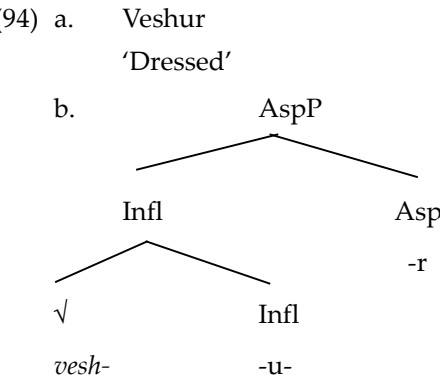

For adjectival participles, which are characterised by the presence of the article, Manzini and Savoia (2018, p. 290) assume the structure in (96). The article that precedes the participle is analysed as a D element (a *linker* in Manzini and Savoia's terms). When the participle realises an adjectival form, it is embedded under the D linker:

(95)  I   veshur

<span style="font-variant: small-caps;">MSG</span> dressed

'Dressed'

(96)  [D i ] [Asp veshur]

For the combination of the participle with the auxiliary *jam* 'be' (in reflexives, middles, unaccusatives, and passives), Manzini and Savoia (2018, p. 278) assume a reduced structure that does not involve the Asp/Event/Voice layer. This is the reason why it is impossible to have a DP external argument. In this case, the external argument surfaces as an oblique. The example the authors use is from the dialect of Shkodër, in Northern Albania:

(97)  a.  Jan   mlu   prej  s      am-s
          be.3PL covered by  FSG.ABL mother-FSG.ABL
          'They have been covered (by mother)'

    b.  Jan [[<sub>VP</sub> mlu DP]  [<sub>PP</sub> prej s ams]]

In combination with the auxiliary *kam* 'have' (in active perfects), Manzini and Savoia (2007, 2018) maintain the same structure even if active perfects have a nominative subject. The idea of the authors is that *kam* results from the incorporation of an oblique preposition into *jam* 'be', à la Kayne (2000). The possessor raises out of the participial clause:

(98)  a.  E      kan    mlu
          him.CL have.3PL covered
          'They have covered him/her'

    b.  DP kan [[<sub>PP</sub> P  ~~DP~~]  [<sub>VP</sub> mlu ]]

If we consider the characteristics of the stative contexts in the Arbëresh variety presented here, we can see that there is solid evidence for a verbal structure: statives support adverbs like *ankora* 'still' and *sempre* 'always' and instrumental PPs. Since adverbs and instrumental PPs adjoin to VP, I conclude that Arbëresh statives involve a VP projection, necessary to license these elements. At the same time, the fact that statives do not permit agentive by-phrases is a clear indication that statives lack a VoiceP projection, where the agentive PP generates. Therefore, I assume that the adjectival head, in statives, embeds a VP and not a VoiceP. Furthermore, given that statives contain pre-articulated adjectival forms, their structure must also include a projection hosting the article, a D head.

(99)

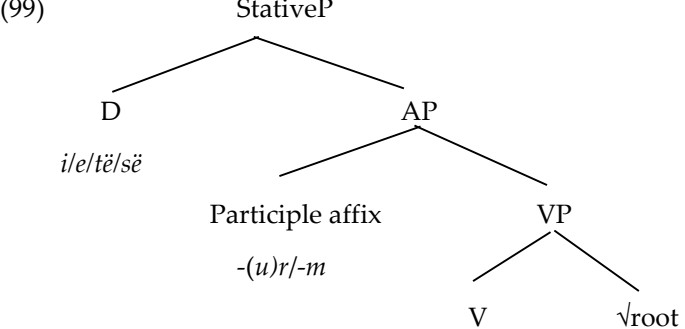

As for the structure of the Arbëresh stative sentences, I will assume they involve a small clause[23] hosting both the DP and the stative adjective, and that the adjectival participle embeds a VP, as is shown by the position of adverbial and PP modifiers.

(100) a.  Der-a      ish        e çar    dje
          door-FSG.DEF be.3SG.IMPF FSG broken yesterday
          'The door was broken yesterday'

    b.  Ghiter-a    ish        e firmarm-e me laps-in
          letter-FSG.DEF be.3SG.IMPF FSG signed-FSG with pencil-ACC.DEF
          'The letter was signed with the pencil'

(101)

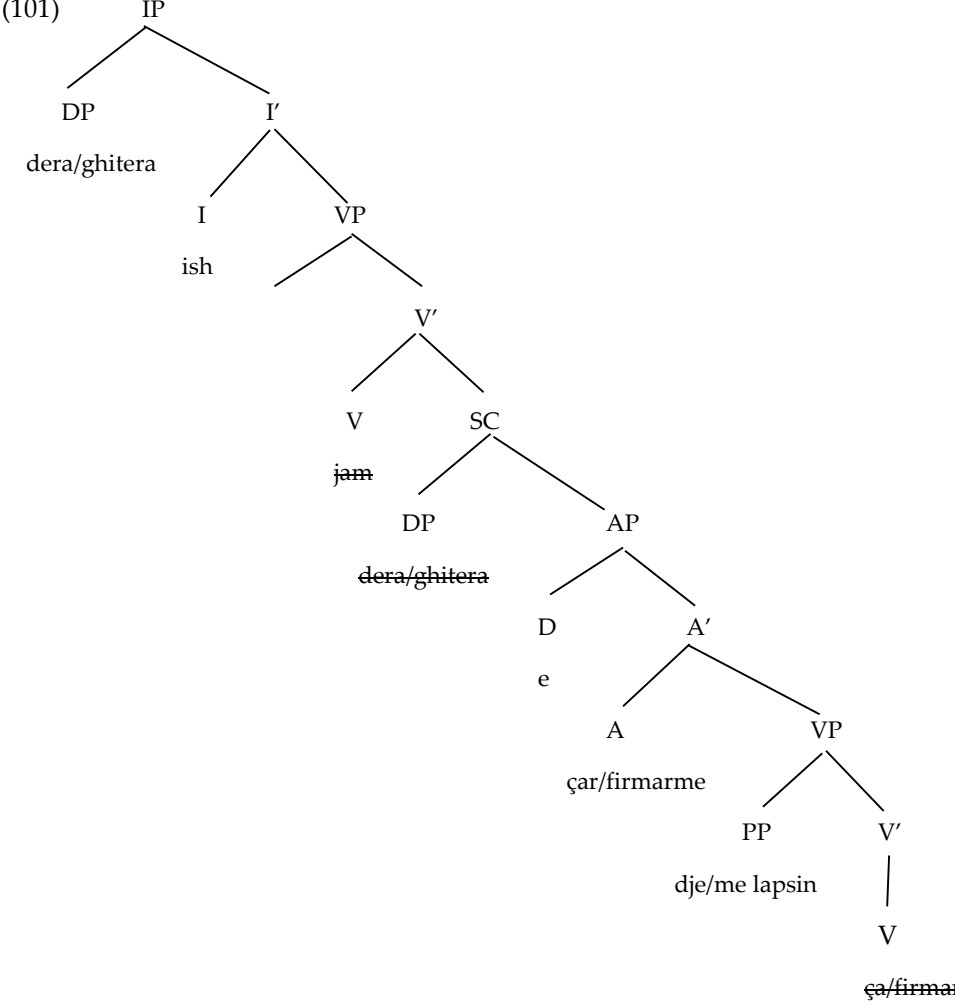

The same structure is assigned to Albanian stative contexts.

Verbal participles, instead, support agentive PPs, as is shown by the following example taken from standard Albanian:

(102)  Hajdut-i     është   arrestuar nga polic-i
thief-NOM.DEF be.3SG arrested by policeman-DEF
'The thief was arrested by the police'

These elements clearly involve a VoiceP projection, where the external argument can be merged.

(103)

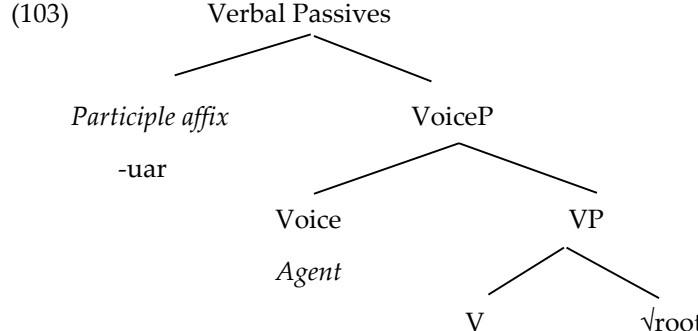

**7. Eventive Passives in Albanian, in Arbëresh, and in Italian**

As we saw in Section 5, the Arbëresh variety of S. Nicola dell'Alto has two types of participles: (a) verbal uninflected participles that appear in active perfects, in combination with the auxiliary *kam* 'have' (104), or in reflexives (105a), middles (105b), and unaccusatives (105c), in combination with the auxiliary *jam* 'be'.

(104)   Maria ka   **hapur** der-in
        Mary have.3SG opened door-FSG.ACC.DEF
        'Mary opened the door'

(105) a.   Maria     osht **larë**
           Mary.NOM be.3SG washed
           'Mary washed'
      b.   Te ai  ristorant osht **ngron** mirë
           at that restaurant be.3SG eaten well
           'In that restaurant we ate well'
      c.   Der-a           osht  **hapur**
           door-NOM.DEF  be.3SG opened
           'The door opened'

(b) Adjectival pre-articulated participles appear in stative contexts:

(106) a.   Der-a       osht   vjetur **e hapur**
           door-NOM.DEF be.3SG remained  FSG opened
           'The door remained opened'
      b.   *Dera osht vjetur **hapur**

There is another context using adjectival pre-articulated participles: passive sentences. Before moving on to the Arbëresh passives, I will present the strategies with which passive sentences are built up in standard Albanian.

In Albanian, passive sentences are formed with a non-active morphology, which is the same that we find in reflexives, anticausatives, and middles. It can be realised in three different ways:

(1) by means of inflectional affixes attached to the verb, in the present and in the imperfect of the indicative, in subjunctive and conditional moods. This strategy is exemplified in (107), which contains the indicative present and imperfect tenses.

(107)   Vajz-a      la-**het**          /lah-**ej**          nga motr-a
        girl-NOM.DEF wash-3SG.PRS.NACT /wash-3SG.IMPF.NACT by sister-DEF
        'The girl is/was washed by her sister'

(2) by means of the invariable clitic *u* in the aorist, in the admirative present and perfect tenses, in the optative present tense, in imperatives, gerunds, and infinitives. This second strategy is exemplified by the aorist in (108). When the clitic *u* is used, the verb has the same person inflection as the active voice.

(108)   Vajz-a      **u**   la          nga motr-a
        girl-NOM.DEF NACT wash.3SG.AOR by sister-DEF
        'The girl was washed by her sister'

(3) with the auxiliary *jam* 'be' followed by the past participle in compound tenses. This strategy is exemplified in (109).

(109)   Vajz-a      është    /ishte    /qe        larë   nga motr-a
        girl-NOM.DEF be.3SG.PRS/be.3SG.IMPF/ be.3SG.AOR washed by sister-DEF
        'The girl is/was washed by her sister'

What is interesting here is the structure in (109), built up with a simple perfect, i.e., a single instance of the auxiliary *jam* 'be' (in the present, in the imperfect, or in the aorist), combined with a participle that is not inflected. This is shown in the following examples.

(110) a.    Vajz-a      është   /ishte   /qe    **larë**   nga motr-a
         girl-FSG.DEF.NOM be.3SG.PRS/be.3SG.IMPF/be.3SG.AOR washed by sister-DEF
         'The girl is/was washed by her sister'

       b.    Vajz-a-t       janë   /ishin   /qenë   **larë**  nga motr-a
         girl-FPL-DEF.NOM be.3PL.PRS/be.3PL.IMPF/be.3PL.AOR washed by sister-DEF
         'The girls are/were washed by their sister'

       c.    Djal-i       është   /ishte   /qe    **lar**ë   nga motr-a
         boy-MSG.DEF.NOM be.3SG.PRS/be.3SG.IMPF/be.3SG.AOR washed by sister-DEF
         'The boy is/was washed by his sister'

       d.    Djem-të      janë   /ishin   /qenë   **larë**  nga motr-a
         boy.MPL-DEF.NOM be.3PL.PRS/be.3PL.IMPF/be.3PL.AOR washed by sister-DEF
         'The boys are/were washed by their sister'

The Arbëresh variety of S. Nicola dell'Alto has maintained the three strategies for the realisation of the non-active morphology: it uses inflectional affixes (111a), the reflexive clitic *u*[24] (111b), and the auxiliary *jam* 'be' + the participle (111c). I exemplify the three strategies with a reflexive construction.

(111) a.    Maria lah-et
         Mary wash-3SG.PRS.NACT
         'Mary washes'

       b.    Maria u   la-ftë
         Mary   NACT wash-3SG.OPT
         'May Mary wash herself'

       c.    Maria osht   larë
         Mary be.3SG.PRS washed
         'Mary washed'

In spite of this morphological similarity, the Arbëresh variety presented here differs from standard Albanian since the non-active morphology, in this dialect, is licit in reflexives (112a), anticausatives (112b), and middles (112c), but not in passives (112d).

(112) a.    Maria la-het          /osht lar
         Mary  wash-3SG.PRS.NACT/be.3SG washed
         'Mary washes/washed'

       b.    Der-a   ha-pet        /osht  hapur
         door-DEF open-3SG.PRS.NACT /be.3SG opened
         'The door opens/opened'

       c.    Te ai ristorant ha-het       /osht ngron mirë
         in that restaurant eat-3SG.PRS.NACT /be.3SG eaten well
         'In that restaurant we ate well'

       d.    *Maria la-het      /osht lar   ka joma
         Mary  wash-3SG.PRS.NACT/be.3SG washed by mother.the
         'Mary is/was washed by her mother'

As is shown in (112d), Arbëresh passive sentences cannot be constructed with morphological affixes or even with a single form of the auxiliary *jam* 'be' followed by the participle. Notice that in no Arbëresh dialect can the passive be formed with morphological affixes or by means of the invariable clitic *u*. The majority of dialects form the passive with the auxiliary *jam* 'be' + a participle (Turano 2023). This is shown by the examples in (113) from the variety of S. Demetrio Corone (Calabria):

(113) a.    *Mesh-a thu-het     ka Peshk-u
         mass-DEF say-3SG.PRS.NAC. by bishop-DEF
         'The mass is said by the bishop'

       b.    *Mesh-a u  tha     ka Peshk-u
         mass-DEF NACT say.3SG.AOR by bishop-DEF
         'The mass was said by the bishop'

       c.    Tribunal-i qe     ngrëjtur ka popull-i
         court-DEF is.3SG.AOR built   by people-DEF
         'The court was built by the people'

In the Arbëresh variety of S. Nicola dell'Alto, the only way to form passive sentences is by using analytical structures built up with a double compound perfect, containing two auxiliaries: a finite form of the auxiliary *kam* 'have'[25] + the participle of the auxiliary *jam* 'be' + the participle of the lexical verb. These forms correspond to a pluperfect like *has/had been washed*.

(114)  a.  Maria ka          /kish        qon e  lar    ka jom-a
           Mary have.3SG.PRS /have.3SG.IMPF been FSG washed by mother-DEF
           'Mary has/had been washed by her mother'
       b.  *Maria osht     /ish      lar   ka jom-a
           Mary  be.3SG.PRS/be.3SG.IMPF washed by mother-DEF
           'Mary has/had been washed by her mother'

This passive configuration, which is different from that of standard Albanian, is modelled on the equivalent Italian constructions. In Italian, too, passive sentences are built up with a double compound perfect, containing two auxiliaries: a finite form of the auxiliary *essere* 'be' + the past participle of the auxiliary *essere* 'be' (in an inflected form) + the participle of the lexical verb.

(115)  a.  Maria è     stat-a    lavat-a    dalla   madre
           Mary be.3SG been-FSG washed-FSG by+the mother
           'Mary has been washed by her mother'
       b.  *Maria è     lavat-a    dalla   madre[26]
           Mary  be.3SG washed-FSG by+the mother
           'Mary is washed by her mother'

Another interesting fact that characterises this variety of Arbëresh is the obligatory agreement on the past participle of the lexical verb[27].

(116)  a.  Vajz-a        ka            /kish          qon **e  lar**    ka jom-a
           girl-FSG.DEF have.3SG.PRS/have.3SG.IMPF  been FSG washed by mother-DEF
           'The girl is/was washed by her mother'
       b.  Djal-i        ka            /kish          qon **i  lar**    ka jom-a
           boy-MSG.DEF have.3SG.PRS/have.3SG.IMPF been MSG washed by mother-DEF
           'The boy is/was washed by his mother'
       c.  Vajz-a-t      kanë          /kishin        qon **të  lar-a**    ka jom-a
           girl-FPL-DEF have.3PL.PRS/have.3PL.IMPF  been FPL washed-FPL by mother-DEF
           'The girls are/were washed by their mother'
       d.  Djem-të      kanë          /kishin        qon **të  lar**    ka jom-a
           boy.MPL-DEF have.3PL.PRS/have.3PL.IMPF been MPL washed by mother-DEF
           'The boys are/were washed by their mother'

Examples in which the participle is uninflected are ruled out.

(117)  a.  *Vajza ka/kish qon **lar** ka joma
       b.  *Djali ka/kish qon **lar** ka joma
       c.  *Vajzat kanë/kishin qon **lar** ka joma
       d.  *Djemtë kanë/kishin qon **lar** ka joma

A similar characterisation is found in Italian. Agreement between the participle and the subject is obligatory, as shown by the contrast between (118) and (119):

(118) a.     La ragazz-a è    stat-a    lavat-**a**     dalla madre
            the girl-FSG   be.3SG been-FSG washed-FSG by+the mother
            'The girl has been washed by her mother'

       b.     Il ragazz-o è     stat-o    lavat-**o**     dalla madre
            the boy-MSG   be.3SG been-MSG washed-MSG by+the mother
            'The boy has been washed by his mother'

       c.     Le ragazz-e sono stat-e      lavat-**e**    dalla madre
            the girl-FPL   be.3PL   been-FPL washed-FPL by+the mother
            'The girls have been washed by their mother'

       d.     I ragazz-i sono stat-i      lavat-**i**     dalla   madre
            the boy-MPL   be.3PL   been-MPL washed-MPL by+the mother
            'The boys have been washed by their mother'

(119) a.     *La ragazza è stato lava**to** dalla madre
       b.     *Le ragazze sono stato lava**to** dalla madre
       c.     *I ragazzi sono stato lava**to** dalla madre

Thus, the Arbëresh examples in (116) contrast with the Albanian ones in (110), since the agreement in Arbëresh is obligatory even though the participle has verbal properties, as it is in eventive passive structures. In Arbëresh, both stative and eventive passive participles have the morphology of adjectives. Both statives and eventives contain inflected participles.

A closer look into the properties of Arbëresh participles shows that the main difference between statives and eventives is not only related to the verbal or adjectival nature of the participle. The difference between the two interpretations also depends on the verbal tense that can occur in these constructions: with a simple perfect, the sentences can only be interpreted as statives (120).

(120) a.     Der-a           **osht**       e çar
            door-FSG.DEF.NOM be.3SG.PRS FSG broken
            'The door is broken'

       b.     Der-a           **ish**        e çar
            door-FSG.DEF.NOM be.3SG.IMPF FSG broken
            'The door was broken'

Statives require a single instance of the finite auxiliary *jam* 'be' + an inflected adjectival participle. The uses of a simple perfect in (120) express a present or past state. The present in (a) leads to the interpretation that the state of the door still holds at the moment of the utterance: the door is in a broken state. The past in (120b) means that the state of the door was held at some point in the past but it is probably no longer broken at the moment of speaking.

As statives, the sentences in (120) do not allow for the presence of agentive by-phrases or instrumental PPs.

(121) a.     *Der-a        **osht**   **/ish**      e çar     ka Beni/ka er-a
            door-FSG.DEF.NOM be.3SG.PRS/be.3SG.IMPF FSG broken by Ben/by wind-DEF
            'The door is/was broken by Ben/by the wind'

       b.     *Der-a        **osht**   **/ish**      e çar      me martel-in
            door-FSG.DEF.NOM be.3SG.PRS/be.3SG.IMPF FSG broken with hammer-DEF.ACC
            'The door is/was broken with the hammer'

Sentences with a compound perfect, instead, are interpreted as passives. As passives, they admit the presence of agentive by-phrases or instrumental PPs.

(122) a.     Der-a           **ka**        **qon** e çar     ka Beni/ka er-a
            door-FSG.DEF.NOM have.3SG.PRS been FSG broken by Ben/by wind-DEF
            'The door has been broken by Ben/by the wind'

       b.     Der-a           **ka**        **qon** e çar     me martel-in
            door-FSG.DEF.NOM have.3SG.PRS been FSG broken with hammer-DEF.ACC
            'The door has been broken with the hammer'

The examples in (122) imply termination of the action/event. They are instances of resultative reading. They are perfective in a narrow sense.

However, in (122), the eventive passive reading is forced by the presence of the agentive PP in (a) and by the instrumental PP in (b). With other types of modifiers, the double perfect can also be interpreted as stative. This is the case, for example, for an adverbial like *a long time*.

(123)  Der-a           ka           qon e çar  për shumë mot
       door-FSG.DEF.NOM have.3SG.PRS been FSG broken for much time
       'The door has been broken for a long time'

The temporal adverbial in (123) triggers a stative interpretation: it means that the state of the door was held at some point in the past, but it no longer holds at the moment of speaking.

Similar facts hold for Italian. The sentence in (124a) can only be interpreted as passive, while the sentence in (124b) has a stative interpretation.

(124)  a.  La porta è    stat-a     rott-a      da Ben/dal vento/col martello
           the door be.3SG been-FSG broken-FSG by Ben/by+the wind/with+the hammer
           'The door has been broken by Ben/by the wind/with the hammer'
       b.  La porta è    stat-a     rott-a     per molto tempo
           the door be.3SG been-FSG broken-FSG for much  time
           'The door has been broken for a long time'can

Coming back to Arbëresh, an interesting fact to note is that the class of participles in *-m*, which cannot be used in active sentences, can also be used in passives, suggesting that a stativizing adjectival head is present in eventive passives:

(125)  a.  Shpi-a           ka           qon e pitarm-e ka Beni
           house-FSG.DEF.NOM have.3SG been FSG painted-FSG by Ben
           'The house has been painted by Ben'
       b.  Ghiter-a          ka           qon e firmarm-e ka diretor-i
           letter-FSG.DEF.NOM have.3SG been FSG signed-FSG by director-DEF
           'The letter has been signed by the director'

In summary, we have seen that eventives like (122) differ from statives by two points: (a) they allow a by-phase (*ka Beni* 'by Ben') and (b) the DP *dera* 'the door' is merged as the complement of the verb, inside the VP. These two elements, plus the presence of adverbial modifiers like *dje* 'yesterday' and *vjet* 'last year', which refer to the time at which the event occurs, and the presence of instrumental PPs like *me martelin* 'with the hammer', clearly indicate that a verbal component is present in these structures. In addition to a VP layer, eventive passives also contain a VoiceP projection where the by-phrase is merged. Moreover, I assume that in eventive passives, AP is also projected (remember that eventive passives are interpreted as statives once we add a temporal adverbial (cfr. (123)). Therefore, in eventive passives, the AP embeds a VP-shell embedded under a VoiceP projection.

(126) IP

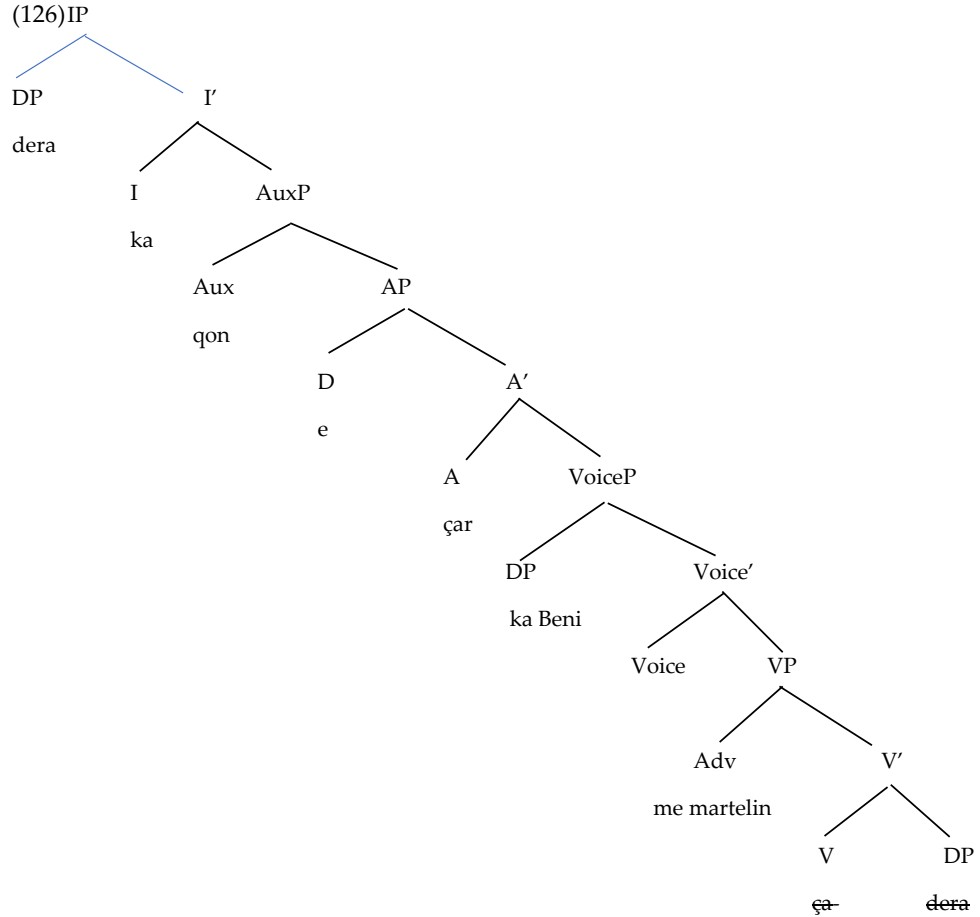

The agreement on the participle can be considered as a Spec-Head agreement activated by the direct object *dera*, which, moving to the subject position, passes through the specifier of the AP projection. This process is similar to the past participle agreement found in Romance languages (Kayne 1989; Belletti 2017).

In summary, Arbëresh resembles Albanian in having verbal participles in reflexives, anticausatives, and middles but it is different from Albanian in passives, since Arbëresh passives require adjectival participles, like in Italian. In the Arbëresh dialect of S. Nicola dell'Alto, the syntax of passive structures has been remodelled so that it also matches Italian in requiring a double compound perfect.

In addition to that, Arbëresh statives, too, are constructed with structures similar to the Italian type, partially different from the Albanian type. The influence of the dominant language is responsible for this contact-induced change on the Arbëresh linguistic system.

**Funding:** This work is part of the project A.L.AR.I.CO, funded by the National Research Program and Projects of Significant National Interest (PRIN–CUP H53D23004040006).

**Institutional Review Board Statement:** Not applicable.

**Informed Consent Statement:** Not applicable.

**Acknowledgments:** I am grateful to two anonymous reviewers for their valuable comments and suggestions.

**Conflicts of Interest:** The author declares no conflict of interest.

**Notes**

[1] An anonymous reviewer asks whether there exist languages with agreement morphology that do not show agreeing participles. I do not know of languages with these characteristics.

[2] The uses and the agreement of the Italian past participles are discussed in detail by Burzio (1986) and Belletti (2017).

[3] The attested Albanian participial forms go back to the Indo-European participles in -\*meno-/\*-mo- and in \*-no-/\*-to-. The forms in *-meno/-mo* have been replaced by the suffixes *-no* and *-to*. Subsequently, the suffix *-no* underwent rhotacism, becoming *-r(ë)*, whereas the suffix *-to* was replaced by *-të*. The participles in *-no* and *-to* were originally adjectives agreeing with the noun in gender, number, and Case (Demiraj 1985).

[4] In this paper, I will not deal with periphrastic verbal forms.

[5] When not specified, the examples are identical in Arbëresh and standard Albanian. The Arbëresh examples are constructed by the author, a native speaker of S. Nicola dell'Alto (Calabria).

[6] In Albanian, the definite article is incorporated with the noun.

[7] According to Kayne (1989), the obligatory agreement activates when the direct object, in its movement to the front of the sentence, it passes through the specifier of the past participle projection.

[8] For discussions on Albanian non-active voice, see Manzini and Savoia (1999, 2008); Kallulli (2006); Kallulli and Trommer (2011); and Manzini et al. (2016).

[9] In Albanian, *by*-phrases can be realised in two different ways: (a) by using the preposition *nga* which selects a nominative DP or (b) by using the preposition *prej*, which selects an ablative DP. The two prepositional phrases have the same meaning and the same distribution.

[10] This is expected since they derive from IE adjectival forms (see note 3).

[11] Genitive, dative, and ablative have the same morphological Case endings. Here, I will use the label OBL (oblique) to refer to them.

[12] Albanian has obligatory *Clitic Doubling* of indirect objects: all dative nouns and pronouns in the argument position must be doubled by the corresponding dative clitic.

[13] The article *e* becomes *të* when the modified noun is an indefinite noun:

    (i)    Takova një djalë të urtë
           meet.1SG a boy-ACC wise
           'I met a wise boy'

[14] The article *e* becomes *të* when the modified noun is an indefinite noun:

    (ii)    Takova një vajz-ë të urtë
           meet.1SG a girl-ACC wise
           'I met a wise girl'

[15] The article *së* becomes *të* when the modified noun is an indefinite noun:

    (i)    I dashë      librin një vajz-e të urtë
           CL give.1SG.AOR book a girl-OBL wise
           'I gave the book to a wise girl'

[16] The article *e* becomes *të* when the modified noun is an indefinite noun:

    (i)    Djem të urtë/Vajza të urta

[17] For our discussion, adjectives without an article are irrelevant, so I do not use them in the examples.

[18] There are exceptions to this statement. Some examples include the following: *i paaftë* 'unable', *i padenjë* 'unworthy', *i paplotë* 'incomplete', and *i paqartë* 'unclear'. I do not know why certain adjectives can be prefixed with the element *pa*, but others cannot. Embick (2004) notes that in English, too, some adjectives allow *un*-prefixation. This is the case, for example, for *unhappy*.

[19] Embick (2004, p. 356) distinguishes two types of stative participles: *resultatives* and *statives*. Resultative participles denote a state resulting from a prior event. Statives describe a simple state.

    (i)    The door was opened    resultative
    [the door was in a state of having become open, i.e., requires state resulting from an event]
    (i)    The door was open    stative
    Here, I will abstract away from these two-way distinctions and simply adopt the traditional distinction between statives and eventives.

[20] The same contrast is found in Italian:

(i)    Maria si è sposata ieri
       'Mary (got) married yesterday'
(ii)   *Maria è sposata ieri
       Mary is married yesterday

[21]  The adverbials *ankora* 'still' and *sempre* 'always' are borrowed from Italian and fully integrated into the Arbëresh grammar. Like other Albanian dialects spoken in Southern Italy, the Arbëresh dialect of S. Nicola dell'Alto has been in contact with Italian for more than five hundred years. Lexical borrowing is the most observable phenomenon that has taken place due to contact.

[22]  Kratzer divides stative participles into two subclasses: target and resultant state participles. Target states are reversible. Resultant states are not reversible. The distinction is mainly semantic, not syntactic.

[23]  On small clauses, see Stowell (1983); Williams (1983); and Moro (1993).

[24]  In the dialect considered here, the aorist has been replaced by the simple perfect, which is the only option for expressing events in the past. However, the presence of the pronominal clitic *u* can be tested with other mood tenses. The example in (111b) contains an optative verbal form.

[25]  In Albanian and Arbëresh, *kam* 'have' is the auxiliary used to form the compound tenses of the verb *jam* 'be':

(i)    Kam    qenë
       have.1SG been
       'I have been'

[26]  This sentence is grammatical with the verb *venire* 'come'.

(i)    Maria viene lavata dalla madre

[27]  Arbëresh dialects show variation in whether or not agreement is attested on the participle. Inflected participles occur, for example, in S. Sofia d'Epiro (Calabria):

(i)    Ai qe        **i bënur** barun ka rregj-i
       he be.3SG.AOR SGM made baron by king-DEF
       'He was proclaimed baron by the king'
Participles without agreement characterise, for example, the dialect of Firmo (Calabria):
(ii)   Qendr-i  qe        **inauguruar** ka dhjak-u
       center-DEF be.3SG.AOR inaugurated by deacon-DEF
       'The center was inaugurated by the deacon'
The dialect of S. Demetrio Corone simultaneously allows for an inflected participle and an uninflected one:
(iii)  Qenë    **ndihur** ka Zoti
       be.3PL.AOR helped by God-DEF
       'They were helped by God'
(iv)   Kriq-i       qe    **i vën** te der-a
       cross-MSG.DEF be.3SG.AOR MSG put at door-DEF
       'The cross was put at the door'

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
