# Peer review of "Stative vs. Eventive Participles in an Arbëresh Variety under the Influence of the Italian Language"

_languages, doi:10.3390/languages9010003_

Round 1

Reviewer 1 Report

Comments and Suggestions for Authors

The article is interesting and well-organized. It depicts a particular condition concerning the Arberesh dialects spoken in Italy, which while preserving the Albanian/ Balkan syntax, however, have been subject to contact with Italian varieties, presenting many lexical borrowings and forms of hybridization in morpho-syntax. The paper focuses on the Arberesh of San Nicola dell'Alto, spoken in Calabria, which, unlike other Arberesh dialects, retains a non-active verbal system very similar to that of Albanian, in which reflexive is realized by the auxiliary jam and no reflexive pronoun occurs, and, what is more crucial, the participle does not agree and is not preceded by the pre-posed article. On the contrary, in the dialect examined by the author, the passive is different from the Albanian one, as it seems to be influenced by the Italian passive, with 'be' and the participle realized in the agreeing form, preceded by the pre-posed article.  The conclusion of the author is that this is due to contact with Italian syntax. A large part of the article discusses the nature of participles, showing that both Arberesh and Albanian have stative forms introduced by the pre-posed article, with the difference that only Arberesh uses these forms in passive A synthetic comparison with the passive in other Arberesh varieties could be interesting, given that also in those the agreeing form of the participle in passive construct emerges, contrasting with other non-active forms, where participles are invariable. Only one observation: in my opinion, the structure of the agreeing participle of passive appears a little complex (on head movement see in particular Chomsky 2021). We have essentially a stative form that, by virtue of its meaning, admits an agentive interpretation and an auxiliary that introduces an eventive reading... Obviously, the cartographic approach makes the solution applied in the paper possible, and in this sense, I have no objection.

Author Response

Thank you for your comments and suggestions. Following your suggestion, I added new examples showing the behaviour of the participle  in other Arberesh varieties. 

The agreement on the participle in passive is the same that Kayne and Belletti use for the participle agreement in Romance languages.

çar sale  a A° e dera salendo passa nello spec A

Reviewer 2 Report

Comments and Suggestions for Authors

This paper is an exploration of stative and eventive participles in the Arbëresh variety of S. Nicola dell’Alto, Calabria, Southern Italy, with a comparison with Standard Albanian.

The paper is descriptive in spirit, and its greatest merit of this paper lies in the wealth of empirical data from a single variety. Very often data from single varieties are fragmented, and this paper contributes to our understanding of the Arbëresh variety of S. Nicola dell’Alto. In this sense this work can function as a foundation for future comparative work in the Albanian varieties of Italy.

The paper also briefly discusses structural proposals by adopting them from previous literature.

I will briefly list some issues I found while reviewing the paper. An elaboration on these issues can be found in the comments to the paper that I attached.

- First, I don't know whether this can only seen in my version of the file, but the example and the trees are severely misaligned, making the reading process very difficult. The author is advised to use Arborwin (Word) of the Forest package (LaTeX) for smooth branch building. As it stands, all branches in all trees need to be fixed. The author should also consider using TAB to align the glosses with the data line in all examples.

- The segmentation in the glosses does not correspond to segmentation in the example line. This should be fixed, otherwise people who are not familiar with Albanian varieties will not be able to appreciate and understand the data. The author should work on amending this issue throughout the paper.

- In those cases where the root can be separated from inflection, - should be used instead of . See: lav-are: wash-INF, but l-e: DEF-F.PL, meaning that the feminine feature cannot be separated from the plural feature, but the lateral can be separated from inflection. The author should check all examples in the paper.

- The choice of upper/lower case in the glosses with respect to gender, number, and case features is not homogeneous. Sometimes it's upper case, sometimes lower case. Here's an example of what I mean:

(26) a. Djali idiot

boy.the.M.SG.NOM idiot.M.SG

‘The idiot boy’

b. Djalin idiot

boy.the.m.sg.acc idiot.m.sg

‘The idiot boy’

These should be made homogeneous throughout the paper.

- Obliques are never correctly rendered in the translation line. Since the translation lacks a preposition, they all seem direct cases. For example, example (26c):

Djalit idiot

boy.the.M.SG.OBL. idiot.M.SG

‘The idiot boy’

The author should fix all obliques in this respect. 

My suggestion is that the paper should be accepted once the author has worked on the aforementioned issues. 

Comments on the Quality of English Language

The English is simple and understandable, however I think that it should be proofread for grammar and style, since it often contains syntactic calques from Italian, for example (p. 14):

Evidence for this distinction comes from the modification.

I have made some comments (attached file).

Author Response

Thank you for your helpful comments and suggestions. I'm sorry for the misalignment of the examples. In the PDF-file, everything is ok, but not in the Word file. I don' know why. 

I revised and arranged the segmentation in all glosses.

All the revisions related to your comments are highlighted in blu. I added footnotes and more information in the text.